# Effects of miR-33 Deficiency on Metabolic and Cardiovascular Diseases: Implications for Therapeutic Intervention

**DOI:** 10.3390/ijms241310777

**Published:** 2023-06-28

**Authors:** Rebeca Ortega, Bo Liu, Shanta J. Persaud

**Affiliations:** Department of Diabetes, School of Cardiovascular and Metabolic Medicine & Sciences, King’s College London, Guy’s Campus, London SE1 1UL, UK; rebeca.ortega@kcl.ac.uk (R.O.); bo.2.liu@kcl.ac.uk (B.L.)

**Keywords:** microRNA, miR-33, lipid metabolism, glucose homeostasis, inflammation, cardiovascular disease

## Abstract

MicroRNAs (miRNAs) are small noncoding RNAs that post-transcriptionally inhibit gene expression. These small molecules are involved in several biological conditions such as inflammation, cell growth and proliferation, and regulation of energy metabolism. In the context of metabolic and cardiovascular diseases, miR-33 is of particular interest as it has been implicated in the regulation of lipid and glucose metabolism. This miRNA is located in introns harboured in the genes encoding sterol regulatory element-binding protein (SREBP)-1 and SREBP-2, which are key transcription factors involved in lipid biosynthesis and cholesterol efflux. This review outlines the role of miR-33 in a range of metabolic and cardiovascular pathologies, such as dyslipidaemia, nonalcoholic fatty liver disease (NAFLD), obesity, diabetes, atherosclerosis, and abdominal aortic aneurysm (AAA), and it provides discussion about the effectiveness of miR-33 deficiency as a possible therapeutic strategy to prevent the development of these diseases.

## 1. Introduction

MicroRNAs (miRNAs) are endogenous noncoding oligoribonucleotides that post-transcriptionally regulate gene expression through their interaction with the 3′ untranslated region (UTR) localised in their target messenger RNAs (mRNAs). This binding affects the stability of these mRNAs, promoting their subsequent degradation or translational repression [1]. Despite their small size (approximately 22 nucleotides), miRNAs have been reported to play important roles in several biological processes due to their highly promiscuous activities. Thus, they have become potential biomarkers for several inflammatory pathologies, including asthma, cancer, and metabolic and cardiovascular diseases. For instance, miR-21 is involved in asthma [2], a chronic inflammatory airway disease, by controlling the survival of human airway smooth muscle cells [3], and miR-17 attenuates inflammation in an animal model of rheumatoid arthritis [4]. Other miRNAs, such as miR-15, miR-16, miR-142, miR-155, and miR-206, play a pivotal role in the development of different types of cancer [5,6,7,8,9], and several miRNAs, including miR-22, miR-372, and miR-122 [10,11,12], are important for the maintenance of glucose homeostasis and appropriate cholesterol levels [13]. In particular, the miR-33 family is crucial in the regulation of lipid metabolism and inflammation [14,15,16], and it has therefore become an important target for the treatment of obesity and dyslipidaemia [17], diabetes [18], and atherosclerosis [19].

MiR-33 is localised in the intronic region of the genes encoding the sterol regulatory element-binding protein (SREBP)-1 and SREBP-2, which are membrane-bound transcription factors involved in lipid homeostasis. Two miR-33 isoforms, known as miR-33a and miR-33b, have been identified in humans and are encoded within *SREBP2* and *SREBP1* genes, respectively. However, only the miR-33a isoform, located in an intron of *Srebp-2*, has been found in mice [20]. The two isoforms share the same seed sequence, but differ by two nucleotides immediately after the seed sequence in their mature forms. Although it has been reported that mRNAs with miR-33 canonical targeting sites are repressed by both synthetic miR-33a and miR-33b, suggesting that they target the same genes, miR-33a and miR-33b have been implicated in having distinct functions in different tissues [21,22,23].

One of the main miR-33 targets is the ATP-binding cassette transporter A1 (ABCA1). ABCA1 actively participates in the reverse cholesterol transport (RCT) process by transferring intracellular cholesterol to high density lipoprotein (HDL) molecules, which drives cholesterol to the liver for its removal [20,24]. MiR-33, which is induced after activation of hepaticfarnesoid X receptor (FXR) [25], represses the expression of liver X receptor (LXR) target genes such as ABCA1 and ABCG1 [26] (Figure 1), thus promoting lipid accumulation in peripheral cells such as macrophages, which contributes to increased size of atherosclerotic plaques [21]. This miRNA also represses the translation of diverse proteins implicated in fatty acid metabolism, including carnitine palmitoyltransferase 1A (CPT1A), hydroxyacyl-CoA-dehydrogenase (HADHB), and carnitine O-octanoyltransferase (CROT) [27]. Impairments in the expression and activity of these proteins can lead to hepatic lipid accumulation, which is of concern in patients with nonalcoholic fatty liver disease (NAFLD) [28].

Under healthy conditions, elevations in plasma glucose after food intake induce insulin secretion from pancreatic β-cells. Insulin reduces blood glucose levels and maintains glucose homeostasis by increasing glucose uptake into adipose tissue (AT) and skeletal muscle, and in the liver, it promotes glycogen synthesis while reducing gluconeogenesis. These effects are achieved by insulin binding to the heterotetrameric insulin receptor, which has intrinsic tyrosine kinase activity [29]. Briefly, insulin binding to its receptor triggers a conformational change that activates the receptor tyrosine kinase, leading to phosphorylation of insulin receptor substrates IRS1 and IRS2 on multiple tyrosine residues. The IRS proteins then act as adapters to recruit signalling proteins containing SH2 domains that recognise phosphotyrosines. The p85 regulatory subunit of phosphoinositide 3-kinase (PI3K) contains two SH2 domains, and its interaction with tyrosine phosphorylated IRS proteins results in activation of the PI3K catalytic subunit and phosphorylation of phosphoinositides on the 3-position of the inositol ring. The phosphoinositide-3,4,5-trisphosphate that is generated recruits the serine/threonine kinases protein kinase B (also known as AKT) and phosphoinositide-dependent kinase-1 (PDK-1) to the plasma membrane via their pleckstrin homology domains, and PDK-1 phosphorylates and activates AKT. Activated AKT promotes translocation of the glucose transporter type 4 (GLUT4) to the plasma membrane of adipocytes and skeletal muscle cells, leading to increased glucose uptake into these cells. AKT activation also promotes glycogen synthase activity, which contributes to glycogen synthesis and storage in muscle cells and hepatocytes [30]. In addition, other proteins, such as the AMP-activated protein kinase (AMPK), are involved in glucose uptake and metabolism. AMPK also enhances GLUT4 translocation into skeletal muscle, and inhibits gluconeogenesis in the liver by repressing the expression of transcription factors responsible for the expression of gluconeogenic enzymes [31]. MiR-33 has been demonstrated to exert important actions on glycaemic regulation, as it inhibits expression of IRS2, decreases levels of phosphorylated AKT, and also inhibits the AMPK subunit-α (AMPKα) [32,33,34], thus compromising glucose uptake into the AT and skeletal muscle and insulin signalling in the liver. Given the involvement of miR-33 in the dysregulation of glucose homeostasis, it is not surprising that recent studies have shown the profound impact of this miRNA on the development of insulin resistance and diabetes [17,23]. 

Several biological processes are regulated by miR-33 since it recognises a wide number of binding sites. It is therefore necessary to understand the molecular mechanism(s) underlying miR-33 post-transcriptional modifications to define whether inhibiting miR-33 signalling may be utilised for the treatment of diverse cardiovascular and metabolic pathologies. For this purpose, different strategies, including genetic miR-33 ablation [17,35] and pharmacological modulation of miR-33 expression using lentivirus or antisense nucleotides [27,36], have been pursued, but further investigations are needed to clarify the beneficial effects of targeting miR-33 on specific organs and tissues.

In this review, we will discuss research in which experimental models of miR-33 repression have been used to define the role of this small RNA in the development of different inflammatory, cardiovascular, and metabolic diseases, and we highlight the gaps existing in miR-33 activity regulation that need to be explored.

## 2. The Role of miR-33 in Dyslipidaemia

Dyslipidaemia is defined as a metabolic disorder characterised by quantitative and qualitative changes in the lipid profile. The main pathological features are increased low density lipoprotein (LDL) cholesterol, a diminished concentration of HDL cholesterol, and hypertriglyceridemia [37]. A key protein involved in this disorder is the transmembrane protein ABCA1, which is responsible for cholesterol homeostasis and plasma membrane remodelling. This transporter transfers intracellular phospholipids and free cholesterol to apolipoprotein A-I (apoA-I) to form nascent HDL particles. Then, the enzyme lecithin-cholesterol acyltransferase (LCAT) transforms free cholesterol into cholesteryl ester to generate mature HDL, which is driven to the liver for biliary secretion or direct fecal excretion [38,39]. 

ABCA1 is highly expressed in hepatocytes [40] and macrophages [41]. However, during dyslipidaemia, an aberrant or low expression of this transmembrane protein is detected in these cell types, thus compromising cholesterol transport [42]. There is compelling evidence that miR-33 regulates cholesterol efflux and HDL biogenesis via repressing ABCA1 expression [43]. Different studies carried out in mice have revealed that miR-33 deficiency or blockade improves ABCA1 expression in peritoneal macrophages and liver and increases plasma HDL cholesterol [20,44,45]. Although these findings are promising, their translational application is compromised given that mice only express the miR-33a isoform, whereas humans express both miR-33a and miR-33b. A miR-33b knock-in (KI) mouse model has therefore been generated, with the aim of analysing the impact of miR-33b on lipid metabolism; these genetically modified mice had decreased ABCA1 protein levels in the liver and peritoneal macrophages, and lower concentrations of plasma HDL cholesterol [46]. These observations have been strengthened by the use of an anti-miR oligonucleotide therapy to suppress miR-33a/b expression in African green monkeys, a model that circumvents the miR-33b deficiency found in mice. Interestingly, pharmacological inhibition of both miR-33 isoforms in the monkeys raised hepatic expression of ABCA1, CROT, and CPT1A, reduced plasma very low density lipoproteins (VLDL), and increased HDL levels [47]. Therefore, the effects observed in monkeys after the inhibitory treatment are consistent with those obtained in the study carried out with the miR-33b KI mouse model. Data derived from both investigations revealed a role for miR-33b in the regulation of cholesterol efflux, and suggests that upregulation of miR-33b is associated with impaired cholesterol levels, and that this effect can be counteracted by inhibiting both miR-33 isoforms. The impact of miR-33 in lipid metabolism has also been analysed in chicken hepatocytes, which showed increased mRNA and protein levels of CROT and HADHB after being transfected with a specific miR-33 inhibitor. Consistent with this, *CROT* and *HADHB* mRNAs were downregulated when miR-33 was overexpressed in the chicken liver [48]. Given that only the miR-33a isoform is expressed in chickens, this study confirms that not only miR-33b, but also miR-33a, contributes to the regulation of cholesterol metabolism. Additionally, several attempts to determine the role of miR-33 in human cells have been pursued. Thus, in vitro assays carried out in human Thp1 macrophages revealed that statin-induced increased expression of miR-33 was accompanied by downregulation of ABCA1 and impaired RCT [49]. Similar findings have been obtained in human hepatic cells transfected with miR-33, which also showed impaired fatty acid oxidation [33,50]. Therefore, miR-33 is emerging as a potential target to treat metabolic diseases such as hypercholesterolaemia [51,52], a disorder characterised by high serum cholesterol levels that compromise the health status of these patients.

## 3. MiR-33 and the Development of Hepatic Steatosis

Nonalcoholic fatty liver disease (NAFLD) is a chronic liver disease characterised by lipid deposition and fat accumulation in the liver in the absence of excessive alcohol intake. This excess of fat storage in the hepatocytes promotes fibrosis and inflammation, two pathological processes that may lead to the development of severe liver diseases such as steatohepatitis, cancer, or cirrhosis [53]. 

It is known that impairments in cholesterol efflux and fatty acid oxidation are associated with NAFLD [54]. Several studies have demonstrated the relationship between miR-33 and NAFLD [55], since miR-33 controls the expression of enzymes such as ABCA1, CROT, CPT1A, and AMPKα that are responsible for maintaining lipid and glucose homeostasis. MiR-33 also represses phosphorylation of IRS-2 and AKT in primary liver cells [34]. This impairment of insulin signalling results in the development of hepatic insulin resistance, a pathological condition strongly associated with hepatic lipid accumulation and NAFLD [56].

Given that high fat diet (HFD)-associated obesity is a common pathological feature of patients with NAFLD [57], feeding mice with an HFD is a common practice to induce NAFLD in rodents [58]. In this way, investigations carried out in HFD-fed mice revealed that genetic deletion of miR-33 promotes hepatic fat accumulation [17] and increases plasma triacylglycerol concentrations [28,59]. These negative effects of miR-33 loss could be explained by upregulation of SREBP-1 in hepatocytes of miR-33 KO mice, since increased SREBP-1 promotes de novo lipogenesis, lipid droplet formation, and triglyceride deposition in the liver of these mice [26,59] (Figure 1). Conversely, other authors note that repression or downregulation of miR-33 for less than 5 months may contribute to improved cholesterol efflux in HFD-fed C57BL/6 mice [60,61], and also to reduced liver inflammation and improved lipid metabolism in apolipoprotein E KO mice subjected to an HFD [62]. In this case, these beneficial outcomes following shorter term miR-33 inhibition, rather than sustained deletion, could be a consequence of enhanced expression of LXR target genes, such as ABCA1, that lead to increased cholesterol efflux [26]. Importantly, SREP2, coexpressed with miR-33, has been shown to inhibit SREBP1 [59]. In this way, when miR-33 is inhibited, fatty acid synthesis is promoted by SREBP1 activation, but cholesterol efflux is improved through ABCA1 derepression (Figure 1). This interaction between SREBP1 and SREBP2 may help to explain the opposite effects regarding lipid homeostasis observed in the liver when miR-33 expression is suppressed or deleted (Figure 1), but it is clear that further research is needed to fully understand the role of miR-33 in regulating lipid homeostasis. 

In addition to an HFD, fructose consumption has also been demonstrated to induce NAFLD [63]. As the intake of added sugars has increased in recent years, several groups of researchers have focused on finding therapeutic targets to prevent fat deposition in the liver, with a focus on miR-33. Some studies have demonstrated that miR-33 expression is decreased in the liver of fructose-fed mice [64,65], while another found increased levels of miR-33 in the hepatic tissue of those mice [66]. However, it was suggested in one of these studies that miR-33 could be released from the liver into the bloodstream in response to hepatic cell damage, as it was found that fructose consumption decreased miR-33 expression in the liver in a time-dependent manner, while it was increased in serum [64]. Thus, miR-33 expression could be increased in hepatocytes of lean mice if there is no liver damage and, if the observations of increased circulating miR-33 in NAFLD are confirmed in further research, it could establish miR-33 as a potential biomarker for NAFLD or hepatic lesion. Glucose is another carbohydrate responsible for lipid accumulation in the liver by promoting fatty acid synthesis [67]. Interestingly, a recent study carried out in geese demonstrated an involvement of miR-33 in this process [68]. The authors found that after goose hepatocytes were exposed to high glucose concentrations, there was increased mRNA expression of *miR-33* and *SREBP2*, the gene that harbours the intronic region of miR-33. In addition, overexpression of miR-33 in glucose-stimulated goose primary liver cells favoured hepatic lipid accumulation, and this effect was counteracted when these cells were treated with a miR-33 inhibitor. Furthermore, miR-33 expression was higher in the fatty liver from overfed geese than normal liver, whereas CROT expression was lower in the first group. Moreover, modulation of miR-33 expression with mimics or inhibitors showed a decrease or increase in CROT expression, respectively. Therefore, this study reveals that glucose-induced fatty liver in geese is mediated by modulatory changes in the miR-33/CROT expression [68].

Given that obesity and diabetes are risk factors that predispose to the development of hepatic steatosis, the effect of exercise on miR-33 expression in the liver of HFD-induced obese and diabetic mice has also been studied. The data suggest that exercise training upregulates miR-33 expression and promotes liver autophagy, which improves hepatic lipogenesis and prevents liver fat accumulation [69]. Conversely, another study carried out in humans revealed that only the miR-33a isoform is increased in morbidly obese patients suffering from nonalcoholic steatohepatitis (NASH), a form of NAFLD that leads to liver inflammation. This increased expression of miR-33a showed an inverse correlation with ABCA1 mRNA and protein levels in these individuals, whereas no changes in miR-33b levels were found between control and NASH subjects [70]. Thus, low ABCA1 protein levels may result in an imbalance in the lipid profile of the obese patients that might contribute to hepatic lipid accumulation. 

A liver-specific miR-33 KO mouse model (LKO) has been generated with the aim of clarifying the role of this miRNA in hepatic tissue. The authors found that LKO HFD-fed mice exhibited a lower grade of hepatic fibrosis and inflammation compared to wild-type (WT) mice [71]. Collagen deposition, macrophage accumulation, and several indicators of liver damage, including alpha smooth muscle actin (αSMA) and alanine aminotransferase (ALT), were reduced after hepatic miR-33 deletion. In addition, in vitro assays carried out in human fetal hepatocytes (L02 cells), in which antagomirs were used to silence miR-33, corroborated that abrogation of miR-33 reduces lipid deposition [61]. 

Therefore, miR-33 inhibition might be a novel therapeutic approach for hepatic steatosis, although further research is needed due to the existing data discrepancy.

## 4. The Role of miR-33 in Adipose Tissue Dysfunction and Obesity

The prevalence of obesity in developed countries has significantly increased in recent years, and it is predicted that one in five adults worldwide will be obese by 2025 [72,73]. 

Obesity is characterised by fat accumulation and AT expansion that is caused by an imbalance between food intake and energy expenditure, with contributions by other genetic, psychological, physiological, social, and environmental factors [74]. AT works as a dynamic immune and endocrine organ that synthesises and secretes a wide number of cytokines, hormones, extracellular matrix compounds, and growth factors that control diverse biological processes [75]. Under obese conditions, adipocytes become dysfunctional and AT suffers quantitative and qualitative changes in its cellular composition, with infiltration of T and B cells, M1 macrophages, neutrophils, and mast cells, which secrete proinflammatory cytokines that promote AT remodelling and insulin resistance [76,77].

It is now apparent that miR-33 may protect against the development of obesity. Thus, mice in which miR-33 had been deleted showed increased weight gain on an HFD compared to control mice, an effect that may be driven by increased food intake due to increased secretion of orexigenic hormones such as ghrelin, or to leptin resistance [17]. These changes in feeding behaviour are likely to be the main cause of increased body weight in miR-33 KO mice, given that no changes in liver SREBP-1 expression, which could alter lipid metabolism, were identified [17]. The same study demonstrated that miR-33 KO mice had increased adipocyte size and macrophage accumulation in white adipose tissue (WAT). These changes in the immune cell profile were also accompanied by elevated levels of the proinflammatory chemokine tumour necrosis factor alpha (TNFα), which is a key mediator of insulin resistance [17]. Unlike miR-33 protecting against proinflammatory events in AT, it has been reported that pharmacological treatment of mice with anti-miR-33 antisense oligonucleotides promotes an anti-inflammatory environment induced by M2 macrophage accumulation in their AT [78]. The increased body weight and insulin resistance of global miR-33 KO mice [17] is in contrast to the beneficial effects of miR-33 inhibition observed in other diseases such as atherosclerosis, where upregulation of miR-33 in plaque macrophages impairs RCT, thus promoting cholesterol accumulation in the aortic wall. Interestingly, targeted deletion of miR-33 in hematopoietic cells in a mouse model of atherosclerosis showed reduced lipid accumulation in atherosclerotic plaques [79]. Therefore, these observations suggest that specific ablation of miR-33 in different organs and cells can avoid the weight gain and glucose dysregulation observed after global miR-33 KO, while positive effects on lipid homeostasis are maintained.

## 5. MiR-33 Impairments in Insulin Resistance and Diabetes

Insulin resistance is defined as the reduced ability of insulin to promote glucose uptake and storage by target organs such as the liver, skeletal muscle, or AT [80]. In the liver, inadequate insulin signalling impairs glycogen synthesis and promotes gluconeogenesis, whereas triglyceride lipolysis is enhanced in the AT, leading to increased glucose production and plasma free fatty acid levels [81]. This metabolic imbalance, when combined with insufficient compensatory insulin secretion, may lead to the development of type 2 diabetes, a pathological condition characterised by hyperglycaemia that can give rise to other comorbidities such as hypertension, neuropathy, retinopathy, nephropathy, and cardiovascular diseases [82]. 

MiR-33 is expressed by islets of Langerhans, and its overexpression in human and mouse islets decreased glucose-stimulated insulin secretion, whereas its inhibition significantly increased insulin release [83]. Further studies carried out in nonhuman primates fed with an HFD revealed that in vivo pharmacological inhibition of miR-33a/b did not influence glucose homeostasis or insulin sensitivity [84]. Conversely, a clinical project revealed that miR-33 levels in serum are increased in prediabetic subjects, but not in control or diabetic individuals, and miR-33 levels were negatively correlated with the homeostatic model assessment for insulin resistance (HOMA-IR). From these observations, the authors suggested miR-33 as a promising biomarker for prediabetes [18]. Further research in insulin-sensitive and insulin-resistant subjects showed that plasma miR-33a and miR-33b levels were influenced by the age of the individuals and by visceral adiposity [85]. Thus, this study revealed that miR-33a was increased in subjects aged between 20 and 39 years compared to individuals aged 40 to 59 years old, whereas miR-33b circulating levels were higher in older, insulin-resistant individuals. Furthermore, an increased concentration of total cholesterol in serum was found when the miR-33a levels decreased in older subjects. Therefore, pharmacological treatments or therapies that selectively enhance miR-33a levels might be useful to counteract the development of obesity-induced insulin resistance, a condition that is aggravated with age. Other researchers have also studied the relationship between miR-33b levels and obesity-induced insulin resistance. One study reported that miR-33b levels were lower in diabetic obese women compared to levels in healthy normal-weight women, and this effect could be reversed by changing dietary habits that lead to reduced weight [86]. Additionally, an analysis of plasma miR-33b levels in patients with diabetic dyslipidaemia and insulin resistance indicated that miR-33b showed a positive correlation with plasma insulin levels and triglycerides, and a negative tendency with HDL cholesterol; these authors suggested that targeting miR-33b using antisense therapy could help to reduce hyperinsulinemia and prevent low HDL levels [87]. Overall, the variety of data available to date demonstrates that the relationship between plasma miR-33 levels and metabolic parameters associated with dyslipidaemia and insulin resistance is not clear, and further research is needed to fully elucidate the role of miR-33 in lipid and glucose metabolism.

Glucagon-like peptide-1 (GLP-1) is a peptide hormone produced by enteroendocrine L-cells of the distal gastrointestinal tract that functions as a key coordinator of glucose homeostasis by enhancing insulin secretion and inhibiting glucagon release, food intake, and gut motility [88]. For these reasons, GLP-1 agonists have been extensively used to treat type 2 diabetes [89]. Recent studies have demonstrated that GLP-1 expression and secretion can be modulated by different miRNAs, including miR-33. For example, it has been demonstrated that the farnesoid X receptor (FXR) enhances miR-33 expression in L-cells and, in turn, miR-33 inhibits expression of target genes such as the cAMP response element binding protein 1 (CREB1) and Gcg, resulting in decreased GLP-1 secretion [90]. Other studies have also established a relationship between miR-33 and GLP-1. Thus, it has been reported that in the pancreas of HFD-fed mice and INS-1 cells, GLP-1 downregulates miR-33 expression, and in INS-1 cells, miR-33 inhibition increases expression of CPT1A and sirtuin 1 (SIRT1), genes involved in fatty acid oxidation [91].

MiR-33 has also been implicated in certain diabetes comorbidities, such as diabetic nephropathy (DN), where it has been observed that patients and rats with DN have elevated circulating miR-33 levels. Additionally, decreased levels of sirtuin 6 (SIRT6) and increased expression of nuclear factor κB (NFκB) were found in the kidneys of these DN rats [92]. The authors suggested that the mechanism by which miR-33 promoted kidney inflammation was through increased NFκB expression secondary to reduced SIRT6, which is a miR-33 target [33] that inactivates NFκB signalling [93].

## 6. MiR-33 Is Involved in Endothelial Dysfunction and Contributes to the Development of Cardiovascular Diseases

The vascular endothelium is a monolayer of endothelial cells that covers the lumen of the blood vessels. It is considered to be an endocrine organ that participates in the regulation of vascular homeostasis. The endothelium has a range of functions, including modulating the traffic of macromolecules between the blood circulation and the vascular wall, regulating the vascular tone, orchestrating the adhesion of mononuclear cells during the inflammatory response, and controlling vasculogenesis and angiogenesis [94,95]. Deregulations, alterations, or aberrant changes in these functions lead to endothelial dysfunction.

A dysfunctional endothelium is characterised by increased vascular permeability, inflammation, and immune system activation [96]. This vascular remodelling enhances the release of proinflammatory cytokines and the expression of several cell adhesion molecules (CAMs) that facilitate leukocyte attachment and migration across the vessel wall to the inflamed tissue [97]. 

Endothelin-1 (ET-1) is a potent vasoconstrictor that has been demonstrated to augment the expression of CAMs, thus contributing to leukocyte recruitment [98]. In certain cardiovascular diseases such as atherosclerosis, the extravasation of monocytes through the endothelium and their subsequent differentiation into M1 macrophages constitutes a key step in the development of atherosclerotic plaques [99]. A recent investigation has shown the involvement of miR-33 in the process of macrophage activation [100]. In this study, ET-1 was overexpressed in a murine model of atherosclerosis, and the authors found that this was associated with upregulation of miR-33 in mouse aortas. In addition, in vitro assays carried out in human umbilical endothelial cells (HUVECs) and bone marrow-derived macrophages (BMDMs) confirmed that ET-1 regulates macrophage polarisation through modulation of the miR-33/NR4A axis. NR4A is a subfamily of orphan nuclear receptors highly expressed in vascular cells under atherosclerotic conditions, given that their expression is induced by inflammatory and proliferative factors such as TNFα, LPS, or IL-1β [101]. Briefly, overexpression of ET-1 in HUVECs stimulated with oxidised LDL (oxLDL) promoted the secretion of miR-33 in exosomes isolated from that medium. When BMDMs were exposed to these exosomes, NR4A expression was reduced [100]. Given that NR4A blocks the inflammatory response by inactivating NFκB [102], M1 macrophage polarisation was promoted. Transfection assays also implicated the ET-1/miR-33/NR4A axis in modulating the expression of proinflammatory mediators [100].

A chronic low-grade inflammation and an oxidative environment are hallmarks of atherosclerosis. During the development of atheroma, recruited monocytes differentiate into macrophages. These tissue macrophages internalise oxidised lipoprotein particles such as oxLDL, and accumulate lipid droplets leading to the formation of foam cells, which secrete proinflammatory cytokines, reactive oxygen species, and matrix-degrading enzymes (metalloproteinases) that contribute to the destabilisation of the atherosclerotic plaque, which increases the risk of thrombotic events that may cause myocardial infarction or stroke [103].

ABCA1 plays a key role in atherosclerosis. This transporter contributes to the reverse transport of cholesterol from macrophages to HDL particles, thus reducing foam cell formation. Then, HDL cholesterol is delivered to the liver for excretion [104]. Several studies have demonstrated a role for miR-33 in cholesterol efflux. It has been reported that mice treated with chitosan nanoparticles containing miR-33 exhibited a reduced reverse cholesterol transport to the plasma, liver, and feces. Additionally, in vitro assays confirmed that miR-33 delivery to naïve macrophages decreased the expression of ABCA1 [105]. In agreement with these observations, it has been reported that upregulation of miR-33a-5p restrained cholesterol efflux and promoted lipid accumulation in THP-1 macrophages by inhibiting ABCA1 and ABCG1 expression, effects that were counteracted using miR-33 antisense therapy [106]. Anti-miR-33 oligonucleotides were also used to demonstrate the beneficial effects that miR-33 inhibition had in the treatment of atherosclerosis. In that study, suppression of miR-33 in LDL receptor KO (*Ldlr*^−/−^) mice reduced the atherosclerotic lesion and lipid content by promoting ABCA1 expression and decreasing the levels of inflammatory markers such as TNFα, toll-like receptor (TLR) 6, and TLR13 in macrophages from anti-miR-33 treated mice. All of these effects were also accompanied by an increase in plasma HDL levels in these mice [19]. Another study also corroborated that miR-33 inhibition attenuated the progression of atherosclerosis in *Ldlr*^−/−^ mice, although this atheroprotective effect was independent of circulating HDL levels [107]. To more deeply analyse the contribution of each miR-33 isoform to the development of the atheroma plaque, the atherosclerotic process has been evaluated in apoE^−/−^ mice expressing miR-33a or miR-33b, both of which are only naturally expressed by humans and other large mammals. It was found that mice expressing miR-33b that were fed with a high fat and high cholesterol diet had an increased atheroma plaque size compared to mice expressing the other isoform. This suggests a greater role for miR-33b in atherogenesis [21]. 

Abdominal aortic aneurysm (AAA) is a cardiovascular disease characterised by a local dilatation of the abdominal aorta. Chronic inflammation of the aortic wall, angiogenesis, and vascular remodelling are its main pathological features. The infiltration of immune cells through a dysfunctional endothelium promotes oxidative stress and induces the expression of inflammatory cytokines and proteases that degrade collagen fibres and elastin, the major components of the extracellular matrix (ECM), thus contributing to vascular smooth muscle cell (VSMC) loss. This weakness of the ECM can lead the aneurysm to rupture [108]. Given that AAA progression is asymptomatic and surgical intervention is the only clinical therapy currently known to be effective to prevent its rupture [109], finding biomarkers for its early detection and new pharmacological targets to prevent AAA development or even its rupture has become a challenge for researchers.

A recent study demonstrated that miR-33-5p expression was increased in human aortas with AAA, whereas ABCA1 expression levels were decreased [110]. Since macrophage infiltration is a key process in the development of AAA, the effect of miR-33-5p downregulation was investigated in human THP-1 macrophages, which revealed that this markedly increased ABCA1 expression and reduced the expression levels of proinflammatory markers such as TNFα, metalloproteinase (MMP)-2, and MMP-9 via the PI3K/AKT signalling pathway [110]. Therefore, suppression of miR-33-5p expression could be a promising therapy to attenuate AAA progression by promoting ABCA1 expression and PI3K/AKT activation. These observations are supported by in vivo studies carried out in animal models of AAA in which miR-33 expression has been suppressed [35,111]. One of these studies demonstrated that the lack of miR-33 improved survival and reduced the size of the AAA lesion. These effects could be explained by the reduced macrophage infiltration and MCP-1 expression detected in aortic cross-sections of miR-33^−/−^ mice. Furthermore, the integrity of the ECM was improved in the aortas of miR-33^−/−^ mice, and this was associated with decreased MMP-9 expression [35]. Studies performed in peritoneal macrophages and VSMCs isolated from miR-33^+/+^ and miR-33^−/−^ mice corroborated these data, with observations of decreased MMP-9 activity in THP-1 macrophages, and a reduced MCP-1 expression in VSMCs from miR-33^−/−^ compared to miR-33^+/+^ mice [35]. Although this research implicates miR-33 in AAA development, the translational outcomes are poor, given that humans have two miR-33 isoforms, miR-33a and miR-33b. To find out if miR-33b is also involved in AAA formation, a recent investigation evaluated the effects of miR-33b blockade in a humanised miR-33b KI mouse model exposed to calcium chloride to induce AAA. It was found that miR-33 KI mice treated with anti-miR-33b nucleotides showed lower macrophage accumulation and reduced expression of proinflammatory markers such as MCP-1 and TNFα, along with decreased MMP activity. Furthermore, anti-miR-33 therapy increased *Abca1* levels and cholesterol metabolism was improved [111]. Thus, suppression of miR-33 expression has been demonstrated to be effective in reducing the formation and development of AAA in preclinical animal models. Therefore, pharmacological inhibition of miR-33 could be a potential therapeutic target to prevent AAA progression in humans.

## 7. Concluding Remarks

Growing evidence supports the key role that miR-33 exerts in regulating glucose homeostasis and lipid metabolism. Thus, in the present review, we have highlighted the importance of miR-33 in the development of metabolic and cardiovascular diseases such as NAFLD, diabetes, atherosclerosis, abdominal aortic aneurysm, and obesity (summarized in Figure 2).

Preclinical studies have shown that upregulation of miR-33 decreases expression of the cholesterol transporter ABCA1 and promotes inflammation. Consequently, cholesterol efflux is compromised and lipid accumulation in specific organs such as liver or inflammatory cells, including monocyte-derived macrophages, is enhanced. These events increase the risk of developing hypercholesterolemia or atherosclerosis. A promising strategy to combat these effects is the suppression of miR-33 expression with anti-sense oligonucleotides. Nevertheless, abrogation of miR-33 expression for an extended period of time can induce hepatic fat deposition, increased adipocyte size, and inflammation in the AT that may lead to hepatic steatosis and obesity.

These various metabolic effects that are regulated by miR-33 may be a consequence of the distinct functional activities in which miR-33a and miR-33b are involved, including regulation of glucose and cholesterol synthesis and insulin signalling [23]. In addition, different expression patterns of these isoforms have been observed in humans, pigs, and genetically modified mice, depending on the tissue [112]. For instance, in human AT, miR-33a is constitutively expressed, whereas miR-33b expression is greatly upregulated during the process of adipocyte differentiation [113]. In the human liver, expression levels of miR-33b are higher than those of miR-33a. Conversely, despite a predominant expression of miR-33b in the liver of miR-33b KI mice, miR-33a is more abundant in the small intestine compared with miR-33b [21]. Additionally, another study revealed that miR-33a expression in AT of pigs was higher than that of miR-33b [22]. Given that the miR-33 isoforms have different expression patterns and may be distinctively regulated in different organs, elucidating the role of miR-33 in certain pathologies is difficult. For this reason, we suggest carrying out future in vitro and in vivo investigations focused on tissue-specific miR-33 deletion to better understand the role of this miRNA in metabolic, cardiovascular, and inflammatory diseases. Although previous studies have demonstrated that specific liver deletion of miR-33 in mice exerts beneficial effects on lipid metabolism, cholesterol efflux, and hepatic fibrosis and inflammation [71], further research is needed to explain the mechanism(s) underlying other pathologies such as diabetes. Deletion of miR-33 in pancreatic beta cells or enteroendocrine cells could be a potential tool to clarify the role of miR-33 in glucose metabolism, insulin secretion, and the inflammatory response, given that functional analysis using ingenuity pathway analysis (IPA) has also predicted a role for miR-33 in immune cell development and inflammation in the small intestine [114].

Recent studies have also shown the contribution of miR-33 in regulating cell proliferation [115], and miR-33 has been suggested to be a tumour suppressor in several cancer cell lines. For instance, increased expression of miR-33 in human hepatoma and pulmonary adenocarcinoma cell lines reduced cell cycle progression [116], and overexpression of miR-33a in a human melanoma cell line inhibited proliferation, invasion, and metastasis after subcutaneous tumour implantation in nude mice [117]. In addition, miR-33 suppressed tumour progression in murine breast cancer cells by contributing to M1 macrophage polarisation [118]. However, miR-33a exerts a different action in human gliomas and cardiac fibroblasts, where increased miR-33a levels contribute to glioma growth [115], and fibrosis and proliferation in human cardiac fibroblasts [119]. This wide variety of responses, depending on the cell type and cellular environment, reinforces the importance of developing new strategies that modulate miR-33 expression in specific tissues and organs. For example, antisense oligonucleotides, miRNA sponges, liposomes, or viral delivery are some promising therapies that could be used for this purpose [120]. However, caution is needed when applying these therapies, given that their effects can be difficult to predict because of the multiple targets of miRNAs. For instance, as we have discussed earlier, miR-33 repression can improve or worsen lipid metabolism in the liver depending on the extent and duration of miRNA inhibition [26,59]. In addition, the outcomes obtained using antisense oligonucleotides can differ from those obtained after miRNA deletion in mice. For example, it has been reported that miR-21 antagomir treatment prevented cardiac hypertrophy and fibrosis in rodents with cardiac disease [121], while another report showed that miR-21 KO mice developed cardiac hypertrophy and fibrosis in response to cardiac stress in a similar way to wild-type mice [122]. In addition, antisense therapy for other miRNAs such as miR-132 has shown promising results in patients with heart failure [123]. Treatment of patients twice over 4 weeks with 0.32, 1, 3, and 10 mg/kg CDR132L, a specific antisense oligonucleotide used to inhibit miR-132, led to reduced cardiac fibrosis, and no serious adverse effects were reported in these patients [123]. Inhibition of miR-33 has not yet been tested in humans, but these previous observations of successful outcomes of antisense oligonucleotide-mediated miRNA repression suggest that it is feasible. Suppressing miR-33 expression may be beneficial for the treatment of diverse metabolic and cardiovascular diseases, and nanoparticles [124] could be used to target the delivery of anti-miR-33 treatments to specific organs and tissues to limit adverse effects such as the increased body weight that is observed after global inhibition of miR-33 expression.

In conclusion, miR-33 is involved in several biological processes related to energy metabolism and cell cycle regulation. Impairments in miR-33 expression can seriously impact glucose and lipid regulatory mechanisms or cell proliferation. Given the current lack of consensus on the beneficial effects of miR-33 deficiency for the treatment of cardiometabolic diseases, it is evident that more research is needed to better understand the role of miR-33 in the regulation of glucose and lipid metabolism.

## Figures and Tables

**Figure 1 ijms-24-10777-f001:**
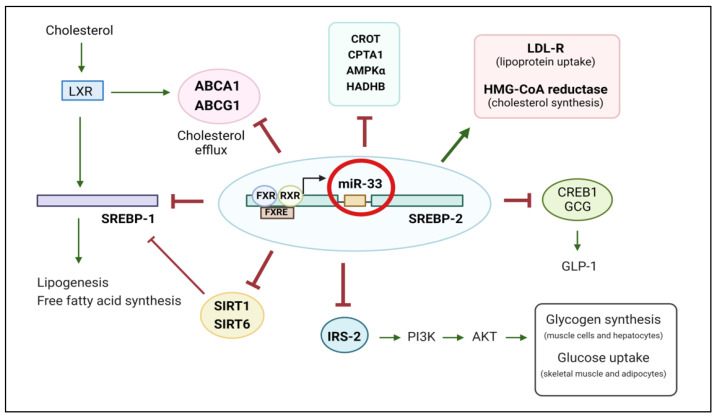
Role of miR-33 target genes in the regulation of glucose and lipid metabolism. MiR-33, which is induced after activation of hepatic FXR in combination with RXR, represses the expression of LXR target genes such as ABCA1 and ABCG1. Consequently, cholesterol efflux from the liver is reduced. MiR-33 also inhibits expression of several proteins implicated in fatty acid metabolism, including CROT, CPT1A, AMPKα, and HADHB while activating LDL-R and HMG-CoA reductase, which are involved in lipoprotein uptake and cholesterol synthesis, respectively. In addition, a recently described interaction between SREBP-2 and SREBP-1 in hepatocytes may explain the negative effects of miR-33 suppression in hepatic fat accumulation. SREBP-1 plays a key role in free fatty acid synthesis, so SREBP-1 repression by miR-33 after activation of SREBP-2 reduces lipogenesis. SREBP-1 expression is also repressed by SIRT1 and SIRT6, two targets genes that are inhibited by miR-33. MiR-33 also inhibits expression of genes such as CREB1 and Gcg, resulting in decreased GLP-1 secretion, thus compromising glucose homeostasis. Insulin signalling is also affected by the inhibition of IRS-2 expression by miR-33. Consequently, activation of the PI3K/AKT pathway is reduced, leading to decreased glucose uptake by adipocytes and skeletal muscle, and reduced glycogen synthesis in muscle cells and hepatocytes. Figure created with BioRender.com.

**Figure 2 ijms-24-10777-f002:**
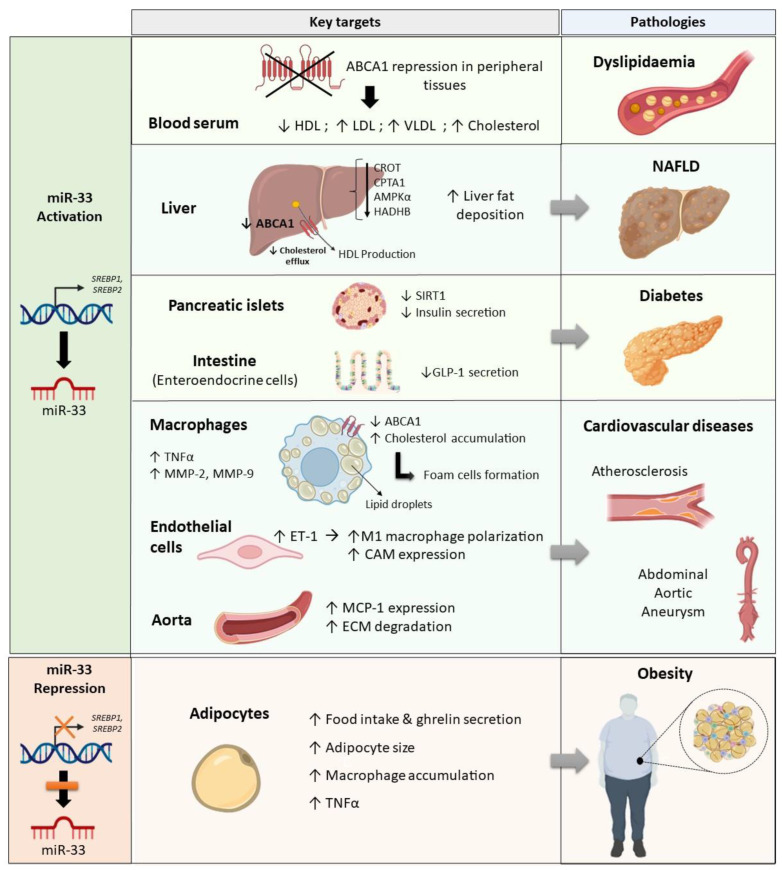
Role of miR-33 in the development of metabolic and cardiovascular diseases. Upregulation of miR-33 has a serious impact on lipid and glucose homeostasis in blood serum and metabolic organs such as the liver, endocrine pancreas, and intestine. MiR-33 is located in introns harboured in the genes encoding sterol regulatory element-binding protein (SREBP)-1 and SREBP-2, which are key transcription factors involved in lipid biosynthesis and cholesterol efflux. Activation of miR-33 represses ABCA1 expression in peripheral tissues leading to dyslipidaemia, a metabolic alteration characterised by decreased levels of HDL particles and increased levels of LDL, VLDL, and cholesterol in blood serum. In the liver, increased miR-33 expression leads to decreased cholesterol efflux and fat accumulation, two factors that may increase the risk of developing NAFLD. In the islets and intestinal enteroendocrine cells, high miR-33 expression results in reduced insulin and GLP-1 secretion, respectively, and the resulting dysregulated glucose homeostasis contributes to the development of type 2 diabetes. In addition, miR-33 has a crucial inflammatory role in macrophages and endothelial cells, in which overexpression of miR-33 impairs lipid metabolism and contributes to cholesterol deposition in the arterial wall. MiR-33 also induces M1 macrophage polarisation and promotes the expression of inflammatory markers such as TNFα, MCP-1, and CAM. As a result, certain cardiovascular diseases such as atherosclerosis and abdominal aortic aneurysm can develop. For this reason, new therapeutic strategies based on the suppression of miR-33 expression seem to be promising for the treatment of these pathologies, but miR-33 repression has a negative impact in the adipose tissue, which may give rise to obesity. Thus, low miR-33 expression promotes food intake and enhances macrophage accumulation and TNFα expression in adipocytes, which also increase in size. Figure created with BioRender.com.

## Data Availability

Not applicable.

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
