# Peer review of "Effects of miR-33 Deficiency on Metabolic and Cardiovascular Diseases: Implications for Therapeutic Intervention"

_ijms, 2023, doi:10.3390/ijms241310777_

Round 1

Reviewer 1 Report

Effects of miR-33 deficiency on metabolic and cardiovascular diseases: implications for therapeutic intervention

This manuscript is a review article dealing with the contribution of miR-33 deficiency with cardiovascular diseases, some modifications were required to improve the quality of the work.

1. Why the authors choose miR-33 in this review?

2. What is the pathogenesis with other cardiovascular diseases?

3. The authors should focus on the miRNA machinery genes responsible for the expression pattern of this miRNA

4. The role of miR-33 should be illustrated in details with obesity, insulin resistance, dyslipidemia with cellular pathways with concerns.

5. The whole manuscript should be revised to avoid grammetical errors.

 Moderate editing of English language required

Author Response

Reviewer 1:

This manuscript is a review article dealing with the contribution of miR-33 deficiency with cardiovascular diseases, some modifications were required to improve the quality of the work.

  1. Why the authors choose miR-33 in this review?

We indicated in the Introduction (page 1, lines 37-40) the reasons for focusing on miR-33 in this review.

  1. What is the pathogenesis with other cardiovascular diseases?

We have focused on the cardiovascular diseases for which there is published evidence of an involvement of miR-33.

  1. The authors should focus on the miRNA machinery genes responsible for the expression pattern of this miRNA

We have now modified the review to consider genes regulating miR-33 expression (see page 2 lines 55-57 and new Figure 1).

  1. The role of miR-33 should be illustrated in details with obesity, insulin resistance, dyslipidemia with cellular pathways with concerns.

The manuscript had already included some signalling pathways in which miR-33 is involved such as signalling by NFκB (pages 7, 8, lines 362-368 and 397-402), a key inflammatory transcription factor associated with diabetic nephropathy and endothelial dysfunction, or the PI3K/AKT pathway which is associated with insulin signalling and the development of AAA (page 9, lines 447-455). We have also provided information on FXR upregulation of miR-33 expression, which results in inhibition of CREB1 expression and GLP-1 secretion (page 7, lines 353-356). We have now included additional information on miR-33 target genes and signalling mechanisms in the liver (see page 4 lines 165-168). We have also added a new figure (Figure 1) that summarises the key signalling pathways downstream of miR-33 in the regulation of glucose and lipid metabolism.

  1. The whole manuscript should be revised to avoid grammetical errors.

We have thoroughly reviewed the revised manuscript to remove any grammatical errors.

Reviewer 2 Report

The submitted manuscript reviews the literature concerning the role of miR-33 in cardiometabolic disease. The review is very well written in its grammar and construction and I have no concerns about the English. I have three minor points that require the authors' attention. If these minor points are addressed this will be a review of some quality.

Author Response

No comments were received.

Reviewer 3 Report

This manuscript outlines the role of miR-33 in a range of metabolic and cardiovascular pathologies and provides discussion about the effectiveness of miR-33 deficiency as a possible therapeutic strategy to prevent the development of these diseases. I still have some concerns.

1.     In Paragraph 4 of the Introduction (lines 63-80, page 2), the authors should provide a brief introduction to the insulin signaling pathway. The detailed information about the role of miR-33 on the insulin signaling pathway in specific diseases should be moved to the relevant section dedicated to discussing the impact of miR-33 on the insulin signaling pathway in those specific diseases.

2.     In Part 2 (lines 105-113, page 3), the authors should first clarify the change in the expression level of ABCA1 in dyslipidemia, and then proceed to discuss how miR-33 can be regulated to impact the expression of ABCA1.

3.     In Part 3 (lines 152-153, page 4), the authors should provide an explanation for how long-term suppression of miR-33 expression promoted hepatic fat accumulation and increased plasma triacylglycerol concentrations. In addition, the authors should compare these long-term effects with the effects of the short-term suppression of miR-33.

4.     In Part 3 (lines 160-163, page 4), the authors should provide an explanation for the observed opposite changes in the levels of miR-33 reported in fructose-fed mice.

5.     In Part 4 (lines 200-217, page 4-5), the authors should streamline the introduction to obesity and emphasize the significance of miR-33 in its development.

6.     In Part 4 (page 4-5), the authors should provide clarification on how the decrease in miR-33 levels contributes to the development of obesity, including the specific target genes and their functional roles. Additionally, it would be insightful to compare these findings with other diseases where miR-33 is upregulated.

7.     In Figure 1, to better compare the role of miR-33 in different diseases, the authors should clearly mark which genes are the direct target genes of miR-33. For genes that are not direct targets, the authors should provide the specific target genes through which miR-33 regulates the expression of these non-target genes.

8.     The authors should provide an explanation of the side effects and potential mechanisms associated with therapeutics targeting the inhibition of miR-33, and propose strategies for enhancing their efficacy and minimizing adverse effects.

English language fine. No issues detected

Author Response

Reviewer 3:

This manuscript outlines the role of miR-33 in a range of metabolic and cardiovascular pathologies and provides discussion about the effectiveness of miR-33 deficiency as a possible therapeutic strategy to prevent the development of these diseases. I still have some concerns.

  1. In Paragraph 4 of the Introduction (lines 63-80, page 2), the authors should provide a brief introduction to the insulin signaling pathway. The detailed information about the role of miR-33 on the insulin signaling pathway in specific diseases should be moved to the relevant section dedicated to discussing the impact of miR-33 on the insulin signaling pathway in those specific diseases.

We have now added a brief introduction to the insulin signalling pathway (page 2, lines 65-83). We have also provided more information on the impact of miR-33 on the insulin signalling pathway in different diseases (page 4, lines 165-168; page 9, lines 447-455).

  1. In Part 2 (lines 105-113, page 3), the authors should first clarify the change in the expression level of ABCA1 in dyslipidemia, and then proceed to discuss how miR-33 can be regulated to impact the expression of ABCA1. Lines 96,97

The change in ABCA1 expression in dyslipidaemia has been clarified on page 3 lines 119-121.

  1. In Part 3 (lines 152-153, page 4), the authors should provide an explanation for how long-term suppression of miR-33 expression promoted hepatic fat accumulation and increased plasma triacylglycerol concentrations (lines 139-143). In addition, the authors should compare these long-term effects with the effects of the short-term suppression of miR-33 (lines 146-149).

We have now discussed possible explanations for the different observations made on hepatic fat accumulation following long-term genetic deletion of miR-33 and its shorter term inhibition (page 4, lines 172-176; page 4, lines 180-189).

  1. In Part 3 (lines 160-163, page 4), the authors should provide an explanation for the observed opposite changes in the levels of miR-33 reported in fructose-fed mice. Lines 154-157

We have expanded this section to provide an explanation (page 4, lines 195-202).

  1. In Part 4 (lines 200-217, page 4-5), the authors should streamline the introduction to obesity and emphasize the significance of miR-33 in its development.

We have now streamlined the introduction to obesity (page 6, lines 275-285) and provided information regarding the increased susceptibility to obesity following global deletion of miR-33 (page 6, lines 286-294).

  1. In Part 4 (page 4-5), the authors should provide clarification on how the decrease in miR-33 levels contributes to the development of obesity, including the specific target genes and their functional roles (lines 209-211). Additionally, it would be insightful to compare these findings with other diseases where miR-33 is upregulated (lines 217-223).

We have expanded Part 4 to consider mechanisms by which miR-33 deletion contributes to the development of obesity (page 6, lines 286-294). Additionally, we have also compared these findings with other diseases where miR-33 is upregulated (page 6, lines 299-307).

  1. In Figure 1, to better compare the role of miR-33 in different diseases, the authors should clearly mark which genes are the direct target genes of miR-33. For genes that are not direct targets, the authors should provide the specific target genes through which miR-33 regulates the expression of these non-target genes.

We agree with the reviewer that easily accessible information on miR-33 gene targets will be helpful in understanding the role of miR-33 in different diseases. We have therefore constructed a new figure (Figure 1) in which we provide information on miR-33 target genes and how they affect other non-direct target genes.

  1. The authors should provide an explanation of the side effects and potential mechanisms associated with therapeutics targeting the inhibition of miR-33, and propose strategies for enhancing their efficacy and minimizing adverse effects.

We have now added information in section 7 (page 13, lines 550-570) related to the use of antisense oligonucleotides to inhibit miRNA expression and we have highlighted that specific deletion of miR-33 in different organs or tissues may be a suitable strategy to selectively ameliorate lipid and glucose dysregulation.

Round 2

Reviewer 1 Report

All modification have been performed.

NA

Reviewer 3 Report

The authors have addressed my concerns.